# Preschool and primary school teachers' attitudes toward students with type 1 diabetes: A cross-sectional study

**María Yehisiri Martín–Báez[1], Candelaria de la Merced Díaz-González** [2]*

1 Insular Maternal and Child University Hospital Complex of Gran Canaria, Canary Health Service, Las Palmas de Gran Canaria, Canary Islands, Spain, 2 Department of Nursing, Faculty of Health Sciences, University of Las Palmas de Gran Canaria, Las Palmas de Gran Canaria, Canary Islands, Spain

* candelaria.diazg@ulpgc.es

## Abstract

### Introduction

Type 1 diabetes mellitus (T1DM) is the most common chronic endocrine disorder in childhood, making teachers key agents in ensuring a safe school environment. The main objective of this study was to evaluate the attitudes and prejudices of teachers at Public Early Childhood and Primary Education Centres (ECPECs) in the municipality of San Bartolomé de Tirajana (SBTGC), on the island of Gran Canaria (Canary Islands, Spain), regarding the care of students with T1DM.

### Materials and methods

A descriptive cross-sectional study was conducted. The target population consisted of 264 teachers from seven ECPECs schools in the municipality of SBTGC, Gran Canaria. Data collection was carried out using the validated instrument *Teacher Negative Attitudes Index toward the Care of Students with Type 1 Diabetes Mellitus (INAPAD-18)*, which allowed for the evaluation of teachers' attitudes and provided an answer to the study's main objective. The research was approved by the Research and Drug Ethics Committee (CEIm) of Las Palmas.

### Results

The final sample consisted of 126 participants, representing a participation rate of 47.72%, which did not reach the size required for statistical representativeness. A total of 15.87% of teachers reported currently having students diagnosed with T1DM. The mean number of years of experience working with students with this condition was 3.77 years [0–35years]. A marked gender disparity was observed, with a predominance of women (84.12%). The mean score obtained on the INAPAD-18 questionnaire was 47.02 (range: 18–90). In this instrument, lower scores indicate more favourable attitudes toward the care of students with T1DM. Although male

**Data availability statement:** All relevant data are within the paper and https://zenodo.org/records/19512736.

**Funding:** The author(s) received no specific funding for this work.

**Competing interests:** The authors have declared that no competing interests exist.

participants showed more favourable attitudes compared to their women, this difference was not statistically significant ($p > 0.05$).

## Discussion/Conclusion

The results indicate generally favourable attitudes, with moderately low INAPAD-18 scores. However, slight deficiencies were observed in teachers' training and perception, consistent with findings from previous similar studies. The ECPEC schools *Juan Grande* and *Las Dunas* stood out for demonstrating the most positive attitudes. It is necessary for educational institutions to implement specific measures aimed at teacher training in order to improve attitudes and ensure appropriate attention to the needs of students with T1DM.

## Introduction

Diabetes mellitus (DM) is a group of chronic metabolic diseases characterized by elevated blood glucose levels resulting from alterations in insulin secretion or action. These alterations affect the metabolism of carbohydrates, lipids, and proteins, leading to multiple long-term complications [1]. According to the American Diabetes Association [2], DM is primarily classified into type 1 (T1DM) and type 2 (T2DM), with T1DM considered the most common chronic endocrine disease in childhood and adolescence. It is caused by the destruction of pancreatic β-cells, culminating in absolute insulin deficiency. The diagnosis is established based on the assessment of biochemical parameters, including elevated fasting blood glucose levels, oral glucose tolerance tests, and glycated haemoglobin percentage, accompanied by characteristic symptoms such as polyuria, polydipsia, and polyphagia. Thus, therapeutic management of T1DM is based on insulin therapy, individualized and balanced meal planning, regular physical activity, and diabetes education—essential pillars that require an interdisciplinary approach [3,4].

Globally, it is estimated that 96,000 children under 15 years of age are diagnosed with T1DM each year [5], while, according to the Spanish Diabetes Society [6], this type of DM represents approximately 10% of all DM cases, with an estimated prevalence of 0.2% (90,000 individuals). Over the past 15 years, the Canary Islands (Spain), and specifically the island of Gran Canaria, have shown an incidence of 30–32 cases per 100,000 children of paediatric age, recording one of the highest rates in Europe [7].

Educational centres are appropriate settings to promote Health Education [8], as well as spaces for socialization and opportunities for personal development that foster self-esteem, critical thinking, and autonomy. In this context, teachers must possess knowledge about T1DM, enabling them to identify signs of decompensation and act immediately in critical situations to ensure safety, protect health, and maintain academic performance [9,10]. Thus, teachers play a fundamental role in preventing both short- and long-term complications associated with DM, while ensuring that students achieve academic success and normal growth and development [11].

Rodrigues et al. [12], in their study, revealed that teacher training through educational programs increases knowledge and confidence in the care of students with T1DM, and highlighted the role of school nurses as professionals who can provide support and ongoing updates. Despite the progress achieved in educational settings, there remains a need to continue exploring this area to identify gaps that may guide the implementation of new strategies.

According to a study published in 2018 [13], few educational centres had glucagon available, and many lacked designated personnel to administer it. However, approximately half of the teachers expressed their willingness to inject it in an emergency, emphasizing the need for specific diabetes training for school staff, as a lack of training or a negative attitude may compromise student safety. Furthermore, hypoglycaemia stands out as the most common complication of T1DM, requiring a rapid and effective response from teachers due to the risk of severe neurological damage, coma, or even death, with the greatest risk observed in younger children [14].

The Government of the Canary Islands (Spain) provides a guide for managing health emergencies in educational centres, which outlines the guidelines and protocols to be followed in response to the most common accidents and emergency situations in schools [15].

Regarding the presence of nurses in Spanish schools, data show that they are present in several types of institutions, although mainly concentrated in centres with students who have special educational needs [16]. According to the Spanish Union of Technical Health Assistants (SATSE) [17], the Canary Islands have only about 22 school nurses, representing a ratio of 1 per 11,000 students. Nationally, the School Nursing Observatory of the General Council of Nursing [18] reports an average ratio of 1 per 6,685 students—far below the recommendations of the U.S. National Association of School Nurses (NASN), which suggests a standard ratio of 1 per 750 students in general school settings, with a recommended reduction to 1 per 225 (2011) or 1 per 125 (2022) in cases involving complex health needs [19,20]. It is necessary to mention that school nursing lacks national legal regulation (non-structural implementation), an aspect that may generate inequalities across different regions of Spain [21]. However, in October 2025, the Master's Degree in Advanced Practice Nursing in School Health was officially published in the Official State Gazette [22].

Similarly, a study conducted in 30 European countries in 2018 showed Norway with the highest ratio (1.4 per 1,000), followed by Finland (1.2 per 1,000) and Iceland (0.9 per 1,000) [23]. In 2018, Díaz and Arias [24] highlighted teachers' demand for the presence of school nurses, considering them essential for promoting healthy habits and managing health incidents, while acknowledging their own lack of necessary training for these responsibilities.

Regarding self-care among students with T1DM, the importance of fostering their progressive participation in the monitoring and management of the disease, according to their age and maturity level, is emphasized [25].

Currently, in Spain, there are six official nursing specialties, including "Family and Community Health" and "Paediatrics." Both specialties encompass specific competencies that qualify nurses to support both students and teachers in educational settings. The growing demand for health care within schools can be met by these specialized nurses, who play a complementary role by guiding teaching staff, families, and students in health promotion, disease prevention, early detection, and health problem management, while also encouraging students' progressive involvement in their own self-care according to their age and maturity level.

During early childhood, students with T1DM lack the autonomy to adequately manage the disease, its treatment, and associated symptoms, requiring supervision from adults (parents and teachers) [26–29]. The time spent in educational centres can exceed five hours per day. Teachers, as directly responsible for supervision during this extensive school period, constitute the key professional group to ensure the safety and adequate care of these students, acting as first responders in emergency situations [29,30].This makes it essential for teachers to possess the knowledge, attitudes, and competencies related to this condition, as these are key factors in both prevention and emergency response. In this regard, assessing teachers' attitudes will help identify strengths and limitations within the educational environment, as well as possible biases or gaps in training that may affect the care, inclusion, and safety of students with T1DM. Such analysis

can provide the educational community with a solid foundation for designing and implementing intervention strategies aimed at improving teacher training and ensuring appropriate school care with the support of healthcare professionals.

Heras-Sevilla et al. (2024) [31] validated the Spanish version of the Inventory of Teachers' Negative Attitudes Toward the Care of Students with Type 1 Diabetes Mellitus (INAPAD), which allows for the identification of negative attitudes and even critical dimensions such as fear, anxiety, teachers' perception of being overburdened, and resistance to integration.

The research question (PICO) [32] was formulated as follows: What are the attitudes and possible biases of teachers in Public Early Childhood and Primary Education Centres (ECPECs) in the municipality of San Bartolomé de Tirajana (SBTGC) toward students with T1DM?

To address this question, the study aimed to evaluate the attitudes and possible biases of teachers in ECPECs within the municipality of SBTGC regarding T1DM, using the validated INAPAD-18 questionnaire.

## Materials and methods

Study Design: An observational, descriptive, cross-sectional study was conducted.

The study population consisted of 264 teachers (N = 264) from public ECPECs in the municipality of San Bartolomé de Tirajana (SBTGC), Canary Islands (Spain). From this population, a representative sample of 157 participants was estimated, calculated with a 95% confidence level and a precision of ±5 percentage points, assuming an expected population proportion of 50%. No loss-to-follow-up rate was estimated [33]. The sample was selected through non-probabilistic convenience sampling.

There were seven ECPECs within the municipality (SBTGC): ECPEC El Tablero, ECPEC Pepe Monagas, ECPEC Alcalde Marcial Franco, ECPEC Las Dunas, ECPEC Oasis de Maspalomas, ECPEC San Fernando de Maspalomas, and ECPEC Juan Grande. The final study sample resulted from participant recruitment conducted over approximately one month (January 10 to February 20, 2025).

Inclusion criteria: active teachers at the time of data collection in any of the seven ECPECs who provided prior informed consent were included. Exclusion criteria: teachers on medical leave, those whose employment contract had ended at the time of receiving the survey link, or those who submitted incomplete questionnaires were excluded.

The validated Spanish questionnaire *Inventory of Teachers' Negative Attitudes Toward the Care of Students with Type 1 Diabetes Mellitus* (INAPAD) developed by Heras-Sevilla et al., 2024 [30], was used with the authors' authorization (email communication with L.A.J.). The 18-item version (INAPAD-18) was applied, which demonstrated a Cronbach's alpha of 0.834.

The INAPAD-18 consists of four attitudinal dimensions (D):

• D1. Teacher self-perception of skills and abilities related to the disease (5 items).

• D2. Attitudes related to the participation and inclusion of students with Type 1 DM (4 items).

• D3. Attitudes regarding the learning and academic performance of students with Type 1 DM (4 items).

• D4. Evaluation of the teacher's professional responsibilities and competencies (5 items).

Each item offered five response options according to a 5-point Likert scale, where 1 = "strongly disagree" and 5 = "strongly agree." According to the questionnaire authors, "a higher score indicates higher levels of unfavourable attitudes toward the care of students with T1DM" [31]. The possible score range is 18–90, categorized into five levels: scores between 18–36 were interpreted as "very favourable attitudes," 37–54 as "favourable attitudes," 55–72 as "unfavourable attitudes," and 73–90 as "very unfavourable attitudes."

The questionnaire included independent variables such as gender, educational centre, teaching experience with students diagnosed with T1DM (in years), type of teaching (early childhood/primary/both), and the presence of students with

T1DM in the teacher's current class (yes/no). On the other hand, the dependent variables corresponded to the 18 items of the INAPAD-18 questionnaire (Table 1).

The recent publication of this tool limits the availability of comparative results from other studies.

Procedure for data collection: It was carried out through "Microsoft Forms" [34]. The link was distributed by the head teacher or head of studies of each school, ensuring that participant identification was not possible. Furthermore, upon accessing the link, teachers were provided with a "participant information sheet" and subsequently given the option to select either "I do not consent" or "I consent". Only those who provided written consent by selecting the latter option were granted access to the INAPAD-18 questionnaire. Fifteen days after the initial distribution, the staff responsible for disseminating the survey link at the centre were asked to remind the faculty members of their opportunity to participate in the questionnaire, and the link was subsequently resent to them.

**Table 1. INAPAD-18 Questionnaire.**

| Items | |
|---|---|
| 1* | The interpretation of the results displayed by a glucometer (blood glucose meter) is straightforward. |
| 2* | As a teacher, I am qualified to care for students with Type 1 Diabetes Mellitus. |
| 3 | It is advisable to prevent students with Type 1 Diabetes from participating in sports activities. |
| 4 | I am concerned about the legal responsibility involved in caring for students with Type 1 Diabetes in the classroom. |
| 5 | It is difficult to attend to students with health problems in the classroom. |
| 6* | I feel capable of administering insulin to students with diabetes. |
| 7 | Caring for students with Type 1 Diabetes exceeds my professional responsibilities. |
| 8* | Students with diabetes can participate in the same extracurricular activities as their peers. |
| 9* | The academic performance of students with diabetes is similar to that of their peers. |
| 10 | Adequate care for students with Type 1 Diabetes requires the presence of a healthcare assistant. |
| 11* | Students with Type 1 Diabetes achieve the same learning outcomes as their peers. |
| 12 | I am unable to respond to emergency situations related to Type 1 Diabetes. |
| 13 | Students with Type 1 Diabetes should attend specialized educational centres. |
| 14 | Students with Type 1 Diabetes have greater difficulty memorizing content. |
| 15 | It is dangerous to have students with health problems in the classroom. |
| 16 | Students with diabetes have greater difficulty concentrating. |
| 17 | I feel uneasy about administering insulin to students with diabetes. |
| 18* | Students with Type 1 Diabetes can play the same games as their peers |

*Items reverse.

For the statistical analysis, the data were first exported to Microsoft version 16 (2025) in Excel format and analysed using the free software JASP version 0.17.11. The analysis was carried out according to: Quantitative variables expressed as mean, range, and standard deviation and, qualitative variables expressed as absolute frequency and percentages.

The association between years of teaching experience and the total INAPAD-18 score was evaluated using Spearman's correlation, after checking normality with the Kolmogorov-Smirnov (KS) test. On the other hand, the relationship between gender and the INAPAD-18 score was analysed using the Mann-Whitney U test, while the relationship between type of ECPEC and INAPAD-18 was analysed with the Kruskal-Wallis test followed by the post hoc test. Statistical significance was considered for $p < 0.05$ in all analyses.

Ethical foundations of the study: authorization was requested from the Department of Education, obtaining a positive response with Registration No. EFPD/72117/2024. Subsequently, this authorization was attached to the application to the Research and Ethics Committee of the province of Las Palmas (CEIm) to carry out the study, and on November 22, 2024, a positive response was received with authorization code 2024-527-1. Based on the project submitted to the CEIm, all participants received the Participant Information Sheet, the Informed Consent, and the Data Protection in a previous section before accepting to participate and accessing the form, respecting the autonomy of the participant and their identity. In addition, a platform contracted by the ULPGC was used as a means for dissemination, storage, and to keep the collected data.

## Results

A sample of 126 participants (47.72%) was obtained, not reaching the estimated sample size; therefore, it cannot be considered representative, limiting the generalizability of the results. The sample consisted of 106 women (84.12%) and 20 men (15.87%), showing a great gender disparity.

Continuing with the quantitative sociodemographic variables, the years of teaching experience with students with T1DM presented a mean (M) of 3.77 years and standard deviation (SD) of 6.25 [0–35]. The high SD relative to the mean indicates substantial inter-individual variability in professional experience. Consequently, it is pertinent to explore, albeit cautiously, whether years of service function as a predictor variable for attitudes toward student diabetes care.

It was observed that most of the respondents (59.52%; n = 75) teach in Primary Education. However, 23.81% (n = 30) of the participants indicated that they work at both educational levels (Early Childhood and Primary Education), while 16.66% (n = 21) specified working exclusively in Early Childhood Education.

Table 2 presents the distribution of teacher responses according to the ECPEC. ECPEC Las Dunas stands out for its high level of participation, accounting for 30.95% (n = 39), compared to ECPEC Juan Grande, which showed the lowest total participation (n = 9; 7.14%). Overall, 73.01% of the participants came from four schools — ECPEC Las Dunas,

**Table 2. Teacher Participation by ECPECs (Public Early Childhood and Primary Education Centres).**

| ECPECs | Frequency | Percentage   Valid | Percentage | Cumulative Percentage |
|---|---|---|---|---|
| Alcalde Marcial F. | 18 | 14.28 | 14.28 | 14.28 |
| El Tablero | 19 | 15.07 | 15.07 | 29.36 |
| Juan Grande | 9 | 7.14 | 7.14 | 36.50 |
| Las Dunas | 39 | 30.95 | 30.95 | 67.46 |
| Oasis de M. | 12 | 9.52 | 9.52 | 76.98 |
| Pepe Monagas | 13 | 10.31 | 10.31 | 87.30 |
| San Fernando M. | 16 | 12.69 | 12.69 | 100.00 |
| Total | 126 | 10.00 | | |

ECPEC El Tablero, ECPEC Alcalde, and ECPEC San Fernando — which suggests that the overall data obtained may tend to reflect more closely the reality of these centres.

In Table 3, the data collected on the presence of students with T1DM in the current classes of the participating teachers are shown. In most schools, the presence of such students in their current classes is low, with percentages below 50%. Notably, in the case of ECPEC Juan Grande, no teacher reported having students with T1DM, whereas ECPEC El Tablero recorded the highest number of teachers with students with T1DM in their classes (n = 8).

Overall, a large number of teachers (n = 106) reported not having students with T1DM in their classes this school year (84.12%), compared to 15.87% of teachers (n = 20) who reported currently having students diagnosed with T1DM. This indicates that most of the surveyed teachers do not currently have direct experience with such students. Therefore, this factor could have an influence depending on the proximity to active cases present in the classroom, since daily contact may be associated with a greater predisposition and level of competence in managing emerging needs.

On the other hand, regarding the results obtained from each participant, the values assigned to each item reflect teachers' perceptions of the care provided to students with T1DM in the classroom. As mentioned earlier, the INAPAD-18 consists of 18 items in total, of which items 1, 2, 6, 8, 9, 11, and 18 are reversed, meaning they reflect favourable attitudes toward the care of students with T1DM. For analysis purposes, it was necessary to recode these items by inverting their values. Conversely, the remaining items retain their original values, assessing less favourable perceptions.

With regard to the items evaluated in their original form (Table 4), the statement with the highest level of agreement concerns the need for the presence of an auxiliary care assistant, with a mean (M) = 4.18 and SD = 1.22, indicating that this view is supported by the majority. In contrast, the item with the lowest level of agreement is the statement suggesting that students with T1DM should attend specialized educational centres, with M = 1.41 and SD = 0.87, which supports the integration of these students in regular educational settings alongside their peers. Additionally, another point to note is teachers' concern about the potential legal responsibility involved, with M = 3.89, as well as their anxiety about insulin

**Table 3. Relationship of Teachers with Students Currently Diagnosed with T1DM.**

| ECPECs | Presence of Students in Their Current Class/Tutorship | Frequency | Percentage Valid | Percentage | Cumulative Percentage |
|---|---|---|---|---|---|
| *Alcalde Marcial* | Yes<br>No<br>Total | 1<br>17<br>18 | 5.55<br>94.44<br>100.00 | 5.55<br>94.44 | 5.55<br>100.00 |
| *El Tablero* | Yes<br>No<br>Total | 8<br>11<br>19 | 42.10<br>57.89<br>100.00 | 42.10<br>57.89 | 42.10<br>100.00 |
| *Juan Grande* | Yes<br>No<br>Total | 0<br>9<br>9 | 0.00<br>100.00<br>100.00 | 0.00<br>100.00 | 0.00<br>100.00 |
| *Las Dunas* | Yes<br>No<br>Total | 5<br>34<br>39 | 12.82<br>87.17<br>100.00 | 12.82<br>87.17 | 12.82<br>100.00 |
| *Oasis de M.* | Yes<br>No<br>Total | 1<br>11<br>12 | 8.33<br>91.66<br>100.00 | 15.38<br>84.61 | 15.38<br>100.00 |
| *Pepe Monagas* | Yes<br>No<br>Total | 2<br>11<br>13 | 15.38<br>84.61<br>100.00 | 15.38<br>84.61 | 15.38<br>100.00 |
| *San Fernando M.* | Yes<br>No<br>Total | 3<br>13<br>16 | 18.75<br>81.25<br>100.00 | 18.75<br>81.25 | 18.75<br>100.00 |

Table 4. Descriptive Analysis of the Items Evaluated in Their Original Form.

| | It is advisable to avoid the participation of students with type 1 diabetes in sports activities. | I am concerned about the legal responsibility involved in caring for students with type 1 diabetes in the classroom. | Elt is difficult to attend to students with health problems in the classroom. | Caring for students with type 1 diabetes exceeds my professional responsibilities | To properly care for students with type 1 diabetes, it is necessary to have the presence of a technical care assistant. | I am unable to respond to emergency situations arising from type 1 diabetes. | Students with type 1 diabetes should attend specialized educational centres | Students with type 1 diabetes have greater difficulties memorizing content. | It is dangerous to have students with health problems in the classroom. | Students with diabetes have greater difficulties concentrating. | I feel uneasy about the idea of administering insulin to students with diabetes |
|---|---|---|---|---|---|---|---|---|---|---|---|
| Valid | 126 | 126 | 126 | 126 | 126 | 126 | 126 | 126 | 126 | 126 | 126 |
| Mean | 1.82 | 3.89 | 3.26 | 3.17 | 4.18 | 3.13 | 1.41 | 1.88 | 2.28 | 2.03 | 3.65 |
| Standard Deviation | 1.06 | 1.28 | 1.25 | 1.45 | 1.22 | 1.21 | 0.87 | 1.15 | 1.19 | 1.14 | 1.32 |
| Minimum | 1.00 | 1.00 | 1.00 | 1.00 | 1.00 | 1.00 | 1.00 | 1.00 | 1.00 | 1.00 | 1.00 |
| Maximum | 5.00 | 5.00 | 5.00 | 5.00 | 5.00 | 5.00 | 5.00 | 5.00 | 5.00 | 5.00 | 5.00 |

administration, with M = 3.65, reflecting a certain level of unease regarding the responsibility placed on teachers. Therefore, it is recommended to provide specialized and appropriate training to strengthen their confidence and ensure optimal care.

Regarding the inversely coded items (Table 5), high scores suggest a favourable attitude, while lower scores indicate less confidence in managing the condition. Thus, in relation to the integration of students with T1DM, the data reflect a low level of agreement regarding participation in the same extracurricular activities (M = 1.71), acquired learning (M = 1.70), academic performance (M = 1.65), and play dynamics compared with other classmates (M = 1.65). In general, since these mean values are low, they highlight a perception that T1DM may significantly affect academic performance and classroom participation.

In the tables presented below, the INAPAD-18 results and subsequent analysis based on each teacher's score are shown. A higher score indicates higher levels of attitudes that are less favourable toward the care of students with T1DM.

Regarding the distribution by gender (Table 6), the high standard deviation value reflects a greater difference in responses, showing variability in attitudes. The minimum score reached 22 points, specifically among women participants, while the maximum scores were 77 points—also in this group—compared to 62 points among men, revealing more positive attitudes toward students with T1DM in the latter group.

Consequently, the INAPAD-18 questionnaire results were analysed by the seven ECPECs schools in the municipality of SBTGC (Table 7). Firstly, the educational centre that obtained the highest mean score was ECPEC Pepe Monagas, with M = 53.23 and SD = 5.73. It was followed by ECPEC Alcalde Marcial, with M = 51.33 and SD = 9.26. Conversely, the school with the lowest score was ECPEC Juan Grande, with M = 40.11 and SD = 10.03, followed by ECPEC Las Dunas, with M = 44.79 and SD = 11.20, and closely by ECPEC El Tablero, with M = 45.15 and SD = 7.96. These differences between centres may be influenced by the small sample size in some ECPECs. In addition, the minimum values were recorded at ECPEC Las Dunas, Juan Grande, and San Fernando, with 22, 23, and 25 points, respectively. Regarding the maximum value, ECPEC Las Dunas stood out with 77 points, followed by ECPEC Oasis and Alcalde Marcial (both with 67 points).

**Table 5. Descriptive Analysis of the Inversely Coded Items (INAPAD-18).**

| | It is easy to interpret the results shown by the glucometer. | As a teacher, I am trained to care for students with type 1 diabetes. | I feel capable of injecting insulin into students with diabetes. | Students with diabetes can participate in the same extracurricular activities as their classmates.. | The academic performance of students with diabetes is similar to that of their classmates | Students with type 1 diabetes achieve the same learning outcomes as their classmates | Students with type 1 diabetes can play the same games as their classmates. |
|---|---|---|---|---|---|---|---|
| *Valid* | 126 | 126 | 126 | 126 | 126 | 126 | 126 |
| *Mean* | 2.28 | 3.66 | 3.58 | 1.71 | 1.65 | 1.70 | 1.65 |
| *Standard Deviation* | 1.08 | 1.27 | 1.43 | 0.88 | 0.91 | 1.05 | 0.91 |
| *Minimum* | 1.00 | 1.00 | 1.00 | 1.00 | 1.00 | 1.00 | 1.00 |
| *Maximum* | 5.00 | 5.00 | 5.00 | 4.00 | 5.00 | 5.00 | 5.00 |

**Table 6. Descriptive analysis of INAPAD-18 Questionnaire Results by Gender.**

| | Women | Men | Overall |
|---|---|---|---|
| *Sample* | 106 | 20 | 126 |
| *Mean* | 47.24 | 45.85 | 47.02 |
| *Standard Deviation* | 10.37 | 11.26 | 10.48 |
| *Minimum* | 22.00 | 25.00 | 22.00 |
| *Maximum* | 77.00 | 62.00 | 77.00 |

**Table 7. Descriptive analysis of INAPAD-18 Questionnaire Results by Educational Centre.**

| ECPECs | Alcalde Marcial Franco | El Tablero | Juan Grande | Las Dunas | Oasis de M. | Pepe Monagas | San Fernando M. |
|---|---|---|---|---|---|---|---|
| *Sample* | 18 | 19 | 9 | 39 | 12 | 13 | 16 |
| *Mean* | 51.33 | 45.15 | 40.11 | 44.79 | 48.83 | 53.23 | 47.31 |
| *Standard Deviation* | 9.26 | 7.96 | 10.03 | 11.20 | 12.02 | 5.73 | 11.59 |
| *Minimum* | 30.00 | 27.00 | 23.00 | 22.00 | 28.00 | 44.00 | 25.00 |
| *Maximum* | 67.00 | 58.00 | 55.00 | 77.00 | 67.00 | 61.00 | 65.00 |

The different averages suggest heterogeneity in perceptions and attitudes, which may be influenced by factors such as type of training, experiences, or established protocols.

Below are the scores obtained by teachers classified according to the presence or absence of students with T1DM in their current class (Table 8). It is mainly observed that teachers who currently have students with the condition in their class (n = 20) obtained lower scores, with M = 44.45 and SD = 10.24, reflecting a better attitude. However, the minimum score shows variation in both groups, with teachers without experience showing low values, indicative of favourable attitudes. Nevertheless, a maximum score of 77 points stands out among teachers without experience, indicating that some participants lack such attitudes. These results could suggest a positive impact—that is, having the experience of having a student with T1DM in the classroom may be associated with a better attitude or predisposition.

**Table 8.  Analysis of INAPAD-18 Questionnaire Results According to the Presence or Absence of Students with T1DM in Their Current Class.**

|  | Experience– Yes | Experience– No |
|---|---|---|
| *Sample* | 20 | 106 |
| *Mean* | 44.45 | 47.50 |
| *Standard Deviation* | 10.24 | 10.50 |
| *Minimum* | 27.00 | 22.00 |
| *Maximum* | 65.00 | 77.00 |

With the aim of identifying the relationship between gender, years of teaching experience, and the ECPEC where teaching activity is carried out, based on the sample and the scores obtained in the INAPAD-18, the following results are presented. To correlate years of teaching experience with students with T1DM and the INAPAD-18 score, a bivariate analysis was performed using Spearman's correlation (Table 9), after confirming the absence of a normal distribution in both variables through the KS test, where years of experience showed KS $p < 0.001$ ($p < 0.05$), while the final questionnaire score presented KS $p = 0.041$ ($p < 0.05$). Since one of the variables did not follow a normal distribution, it was not necessary to assess whether they were paired. The result shows a very weak negative relationship, rho $= -0.117$ (the more years of experience, the more positive the attitudes), but without a statistically significant relationship ($p > 0.05$). In addition, a scatter plot (Fig 1) has been included to illustrate the distribution of the sample in relation to years of experience and scores for negative attitudes, which shows a trend line with a slight downward slope, suggesting that, as experience with these students increases, scores for negative attitudes tend to decrease slightly, although the effect is small.

Weak inverse association between years of teaching experience with students with type 1 diabetes and negative attitude scores (INAPAD-18)

To identify the possible relationship between gender and the INAPAD-18 score, a bivariate analysis was performed using the Mann-Whitney U test (Table 10). As mentioned earlier, the INAPAD-18 score did not follow a normal distribution, and the variables were not paired (measurements were taken at different times). The result shows no significant relationship, 0.711 ($p > 0.05$), between gender and the total score obtained in the INAPAD-18.

On the other hand, an attempt was made to determine whether there are significant differences between the ECPEC to which the teachers belong and the INAPAD-18 score, taking as a reference the school with the lowest mean (ECPEC Juan Grande), which showed the most positive attitudes. Since this is a qualitative polytomous variable, a bivariate analysis was carried out using the Kruskal-Wallis test (Table 11). The result shows $p = 0.007$, indicating that differences exist between the two variables, although not specified. For this reason, a post hoc test was conducted (Table 12) to show the differences between the ECPECs with the most positive attitudes and the other schools.

## Discussion

The objective of this study was to evaluate the prejudices and attitudes of teachers from the ECPECs in the municipality of SBTGC toward T1DM. The findings show the existence of favourable attitudes (M = 47.02), with relatively low scores obtained on the INAPAD-18. As lower scores on this scale indicate more favourable attitudes, these results suggest a generally positive profile, although high values were also observed in certain cases. Since the validated instrument

**Table 9.  Spearman Correlation Between Years of Teaching Experience and INAPAD-18 Score.**

| Variable |  | INAPAD-18 Score |
|---|---|---|
| *Years of Teaching Experience with Students with T1DM* | Spearman´s rho<br>p-value | −0.117<br>0.191 |

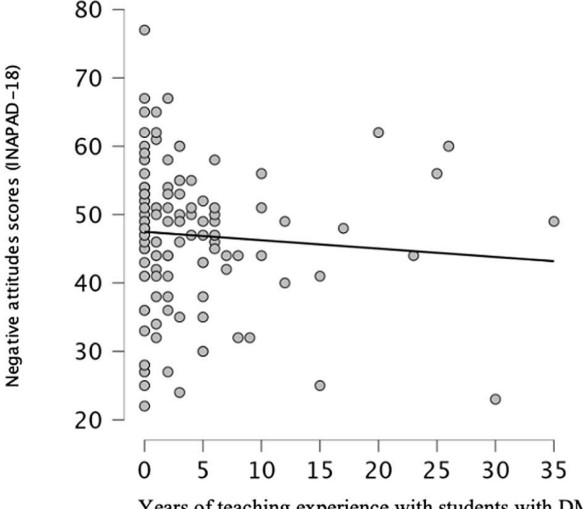

**Fig 1. Relationship between teaching experience and negative attitudes.**

**Table 10. Mann-Whitney U Test Between Gender and INAPAD-18 Score.**

| Independent Samples Test | Mann-Whitney | | |
|---|---|---|---|
| | W | Df | P |
| *INAPAD-18 Score* | 1116.000 | | 0.711 |

**Table 11. Kruskal-Wallis Test Between ECPECs and INAPAD-18 Score.**

| Kruskal-Wallis test | | | |
|---|---|---|---|
| | Statistic | Df | P |
| *ECPECs* | 17.690 | 6 | 0.007 |

**Table 12. Post hoc test.**

| Dunn's Post Hoc Comparisons – ECPECs | | | | |
|---|---|---|---|---|
| ECPECs | Z | Wi | Wj | P |
| Juan Grande – Alcalde Marcial F. | 2.952 | 81.972 | 38.00 | 0.003* |
| Juan Grande – El Tablero | 1.178 | 55.395 | 38.000 | 0.239 |
| Juan Grande – San Fernando | −1.739 | 38.000 | 64.438 | 0.082 |
| Juan Grande – Las Dunas | −1.292 | 38.000 | 55.436 | 0.196 |
| Juan Grande – Oasis de M. | −1.766 | 38.000 | 66.417 | 0.077 |
| Juan Grande – Pepe Monagas | −3.146 | 38.000 | 87.769 | 0.002* |

*$p < 0.01$.

assigns low values to positive attitudes, this reveals slight deficiencies in teachers' training and perception, generating a certain degree of confidence in caring for students with T1DM in educational centres.

The results collected are consistent with previous research that has highlighted the lack of training as a determining factor in teachers' perceptions of diabetes. Carral F. et al. [13] identified that a large proportion (46.9%) of teachers from 44 public schools around the Hospital Universitario Puerto Real stated that they were willing to administer glucagon if necessary, expressing concern about the availability of resources and qualified personnel to manage emergency situations. Therefore, optimal training is essential not only for teachers to gain a better understanding of the needs of children but also to foster positive attitudes toward managing the condition, thereby improving behaviours that could otherwise have a negative impact. This supports the hypothesis that lack of knowledge is one of the main challenges for the inclusion of these students.

Rodrigues et al. [12] reported that teachers who participated in the educational intervention on diabetes showed significant improvement, noting that 78.85% of teachers increased their knowledge, with a statistically significant change ($p < 0.01$) in the scores obtained. Likewise, they gained greater confidence, contributing to improved care and perceived support for students. Similarly, greater confidence would help reduce anxiety in situations requiring their intervention.

Similarly, in 2011, the Government of the Canary Islands launched an educational program titled "Improving the Quality of Life of Schoolchildren with Diabetes in the Canary Islands" [35], aimed at training teachers from public schools belonging to the Network of Health-Promoting Schools in the necessary knowledge about diabetes. The goal of this program was to prepare teachers to respond appropriately to any situation and to support the integration of students with T1DM in the school environment. In its first edition, held in 2012, 82 teachers from 69 public schools received training; however, no specific information has been found regarding the continuity of this program to date. Despite this, the health and education authorities of the Canary Islands have continued to promote and expand similar initiatives, such as the "School Nursing Program of the Canary Islands Health Service". Nevertheless, it must be remembered that school nursing is not structurally established, which may explain low school nurse-to-student ratio (1:11,000) —far below international recommendations (1:750)— and reflects a clearly insufficient provision of school nursing services that may be modulating the results of this study [17,19,20].

On the other hand, as noted by Armas-Junco L. [36], in a study involving 652 teachers from schools in Burgos, 39.7% expressed difficulty in attending to students with health problems in the classroom. Furthermore, 53.3% of teachers expressed concern about the legal responsibility involved in caring for a student with T1DM. This is consistent with the present study, which obtained average INAPAD-18 values ranging between 3.26 and 3.89, showing mild concern and a lack of confidence regarding preparedness to handle various situations related to the condition.

At the same time, a considerable difference in various perceptions and attitudes was found. In the study, high percentages were observed in aspects favouring students' participation in the same extracurricular activities, games, sports, learning, and academic performance. Although overall attitudes were mostly favourable, relevant negative perceptions persisted in specific domains such as participation in activities and academic performance (INAPAD-18 ranged between 1.41 and 1.82). This suggests the need to promote an inclusive environment in these areas, as the teachers' lack of confidence influences exclusive treatment toward these students, which may generate a sense of difference among peers and vulnerability in group integration.

Regarding the analysis of sociodemographic variables, years of teaching experience did not show a statistically significant association with attitudes toward students with T1DM. However, the correlation coefficient rho = − 0.117 indicated a weak negative relationship, suggesting that a higher number of years of experience tends to be associated with more positive attitudes toward these students. Conversely, previous studies such as that of Luque-Vara et al. [37] have suggested that teachers with less than 10 years of experience tend to show favourable attitudes, due to their recent training and motivation to learn. However, experience does not manifest uniformly; it can be expressed in different ways. The seniority

acquired throughout a professional career may promote greater confidence in some, while in others, it may imply a higher sense of personal responsibility.

Nevertheless, although this study provides relevant insights into teachers' prejudices and attitudes regarding the care of students with T1DM, it is essential to highlight its limitations. Despite considerable effort, one of the main challenges of this work was the inability to achieve a representative sample, with a participation rate of 47.72% (n = 126), which limits the generalizability of the results to the selected municipality. Furthermore, potential bias resulting from self-assessment should be considered, as well as differences in teachers' prior training and access to it, and variations among schools— including differences in available resources—which may also influence attitudes and should be taken into account when interpreting the results. Likewise, the data collection period may have negatively influenced participation, as it coincided with the period following the Christmas holidays, as well as with possible coverage of teacher substitutions. These circumstances may have contributed to teachers' reluctance to participate, among which could be found the high workload, the limited confidence in health-related knowledge, the concern over potential evaluation, as well as the professional implications of their possible responses. The perception of insufficient knowledge to manage urgent situations related to diabetes could be explained by the inadequacy or absence of training programs, stemming from the lack of trainers or school nurses responsible for periodically delivering and assessing teachers' knowledge and skills in this context. As noted previously, a structured implementation of school nursing in Spain remains necessary [21].

Among the strategies to enhance teachers' knowledge and confidence in caring for students with diabetes, theoretical training stands out, as certain online programs have demonstrated significant improvements in both knowledge and teacher confidence, with retention maintained up to 12 months [38]. Likewise, evaluating training impact is equally advisable to assess program effectiveness. Authors recommend pre-post evaluations. Thus, a 2026 study [39] found significant increases in knowledge, perceived self-efficacy, and ability to recognize hypoglycemia symptoms (all p < 0.001), similar to virtual pilot studies showing significant improvements in diabetes technology use, basic management, and ketone handling (all p < 0.001) [40]. Finally, simulation emerges as the most robust evidence, with numerous studies demonstrating significantly increased confidence in managing these patients during emergency situations [41–43].

Nevertheless, this study represents one of the first applications of the INAPAD-18, as it was recently validated (2024). Since no previous research has used this tool due to its recent validation, the findings obtained have been aligned with similar studies that analysed comparable barriers in student care. Therefore, it would be appropriate to continue pursuing this line of research through new studies that examine the validity of the instrument across a variety of school settings, including public, subsidised, and private institutions located in different geographical areas. Moreover, conducting longitudinal studies would make it possible to identify changes in teachers' attitudes over time.

Likewise, the scarcity of research devoted specifically to this topic has posed an additional challenge for the discussion and comparison of the findings.

In conclusion, there is a clear need for further complementary research to expand knowledge on this subject and to design strategies aimed at improving the care of students with T1DM in schools.

## Conclusion

The participating teachers showed mostly favourable attitudes toward the care of students with T1DM, according to the results reported using the INAPAD-18 questionnaire. However, significant differences were detected between educational centres, with more positive attitudes in some ECPECs and less favourable ones in others, showing teaching experience as a factor that promotes more positive perceptions. The findings highlight the need to implement specific training programs to strengthen teachers' knowledge, self-confidence, and ability to respond to life-threatening emergencies in the school environment. Furthermore, the importance of increasing the nurse-to-student ratio in schools is emphasized, preferably through regulated specialties such as Family and Community Health, Paediatrics, or the Master's Degree in Advanced Practice Nursing in School Health (recently approved in October 2025). However, its effective deployment

across all autonomous communities will likely require a progressive implementation process and sustained political commitment. This may temporarily perpetuate territorial differences but is essential to ensure comprehensive, safe, and collaborative care. More studies are needed to reflect the real situation of educational centres in the face of life-threatening emergencies, to promote educational programs that include simulations, and to evaluate the impact of these educational interventions to identify gaps and establish institutional foundations for developing coordinated protocols that ensure inclusive and equitable care.

## Acknowledgments

We would like to thank the Department of Education of the Government of the Canary Islands, the principals of the ECPECs in the municipality of San Bartolomé de Tirajana, and the teachers of these schools; without their collaboration, this study would not have been possible. We are also grateful to the authors of the INAPAD instrument for kindly authorizing its use in this study.

## Author contributions

**Conceptualization:** María Yehisiri Martín–Báez.

**Data curation:** Candelaria de la Merced Díaz-González.

**Formal analysis:** Candelaria de la Merced Díaz-González.

**Investigation:** María Yehisiri Martín–Báez.

**Methodology:** María Yehisiri Martín–Báez, Candelaria de la Merced Díaz-González.

**Visualization:** María Yehisiri Martín–Báez, Candelaria de la Merced Díaz-González.

**Writing – original draft:** María Yehisiri Martín–Báez, Candelaria de la Merced Díaz-González.

**Writing – review & editing:** María Yehisiri Martín–Báez, Candelaria de la Merced Díaz-González.

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
