## [Decision Letter · Decision Letter 0]

16 Feb 2026

PONE-D-25-62219Preschool and Primary School Teachers’ Attitudes Toward Students with Type 1 Diabetes: A Cross-Sectional StudyPLOS One

Dear Dr. Díaz-González,

Thank you for submitting your manuscript to PLOS ONE. After careful consideration, we feel that it has merit but does not fully meet PLOS ONE’s publication criteria as it currently stands. Therefore, we invite you to submit a revised version of the manuscript that addresses the points raised during the review process.

We look forward to receiving your revised manuscript.

Kind regards,

Afagh Hassanzadeh Rad

Academic Editor

PLOS One

Journal Requirements:

4. In the online submission form, you indicated that all data will be made available to interested parties upon request to the authors.

5. Please include your tables as part of your main manuscript and remove the individual files. Please note that supplementary tables (should remain/ be uploaded) as separate "supporting information" files.

Reviewers' comments:

Reviewer's Responses to Questions

**Comments to the Author**

1. Is the manuscript technically sound, and do the data support the conclusions?

Reviewer #1: Partly

Reviewer #2: Yes

2. Has the statistical analysis been performed appropriately and rigorously? 

Reviewer #1: I Don't Know

Reviewer #2: Yes

3. Have the authors made all data underlying the findings in their manuscript fully available?

Reviewer #1: Yes

Reviewer #2: Yes

4. Is the manuscript presented in an intelligible fashion and written in standard English?

Reviewer #1: Yes

Reviewer #2: Yes

5. Review Comments to the Author

Reviewer #1: The author indicate why the specific focus on teachers was made in this study.This can make the introduction more stronger .

Non-probability sampling can pose certain limitations in its implementation, as the results may not be fully generalizable to the entire population of teachers. Explain the challenges of this type of sampling and its impact on the final findings.

Further explanation should be given about how the schools were selected, whether they were randomly selected or purposefully selected for a specific reason? Was it random sampling or not?

Are specific statistical analyses considered to compare different groups of teachers (by gender, experience, type of teaching, etc.)?

If possible, it may be useful to mention challenges or time constraints, as research during this time frame may be affected by factors such as holidays or unexpected circumstances (e.g. health or natural crises)

It is important to point out the limitations of the study, especially the low participation rate (47.72%), and the fact that the study results cannot be fully generalized to the entire population is very clear and useful.

Also, pointing out the lack of similar research in this area helps to better understand the existing research problems and the scope for expanding the research.

Explain more about limitations on participation rates: In addition to stating the percentage of participation, you might explain whether factors such as teacher reluctance or time constraints contributed to the low participation. This can help provide more clarity about the reason for the limitations in the results.

In the section on suggestions for future research, it would be useful to mention the possibility of conducting longitudinal studies rather than just cross-sectional ones. This longitudinal study could help identify changes in teachers' attitudes over time.

Perhaps adding details about how this tool could be applied to other schools or different regions could make future research more engaging and scalable.

Also, in the section referring to the nurse-to-student ratio, explaining more about the current situation in schools (what this ratio is and what challenges exist) could make the suggestions more effective.

Explain more about limitations on participation rates: In addition to stating the percentage of participation, you might explain whether factors such as teacher reluctance or time constraints contributed to the low participation. This can help provide more clarity about the reason for the limitations in the results.

In the section on suggestions for future research, it would be useful to mention the possibility of conducting longitudinal studies rather than just cross-sectional ones. This longitudinal study could help identify changes in teachers' attitudes over time.

Perhaps adding details about how this tool could be applied to other schools or different regions could make future research more engaging and scalable.

Also, in the section referring to the nurse-to-student ratio, explaining more about the current situation in schools (what this ratio is and what challenges exist) could make the suggestions more effective.

Explain more about limitations on participation rates: In addition to stating the percentage of participation, you might explain whether factors such as teacher reluctance or time constraints contributed to the low participation. This can help provide more clarity about the reason for the limitations in the results.

In the section on suggestions for future research, it would be useful to mention the possibility of conducting longitudinal studies rather than just cross-sectional ones. This longitudinal study could help identify changes in teachers' attitudes over time.

Perhaps adding details about how this tool could be applied to other schools or different regions could make future research more engaging and scalable.

Also, in the section referring to the nurse-to-student ratio, explaining more about the current situation in schools (what this ratio is and what challenges exist) could make the suggestions more effective.

Mentioning the need for simulations and evaluating educational impacts can guide future research to examine whether these trainings have a positive impact on teacher preparation.

It may be useful to refer to methods for evaluating the impact of training. For example, do teachers score better on subsequent assessments after training? This can strengthen conclusions and emphasize the effectiveness of training programs in particular.

In the case of emergency training and simulations, more details may be provided on how these simulations will be implemented (e.g., use of medical simulation technologies or training workshops)

Reviewer #2: Dear Editor

Thank you for the opportunity to review this manuscript.

My comments are as follows:

Use of Technical Terms: In the introduction and abstract, the use of terms such as INAPAD-18 is appropriate. However, it may be confusing for non-specialist readers. I recommend including a brief explanation of the nature and purpose of this scale to improve clarity.

Data Collection Tool: The INAPAD-18 questionnaire was recently validated (2024). Therefore, comparisons with previous studies are limited, and this should be clearly stated in the manuscript.

Sample and Study Methodology: The sample included 126 participants, and convenience sampling was used. Given the response rate of 47.72% and the fact that the target sample size was not reached, the sample is not fully representative, limiting the generalizability of the results. It is recommended to explain how this limitation may have influenced the findings and what measures were taken to address it.

Results and Data Analysis: The data in the abstract are somewhat raw, and in-depth analysis is limited. For example, the statement that “differences between men and women were not significant” lacks the p-value.

The mean years of teaching experience with students with T1DM was 3.77 years, with a standard deviation of 6.25 and a range of 0–35 years. The high SD indicates substantial heterogeneity in professional experience, and it is recommended to explore whether this variable affects teachers’ attitudes.

Some analyses, such as the effect of teaching experience, could be illustrated with graphs or charts to make the findings clearer.

Study Limitations: Besides the small and non-representative sample and the recent validation of the questionnaire, other limitations—such as self-report bias, prior teacher training, and differences between schools—are not clearly addressed.

Writing and Structure: Some data are repeated in the text and tables, which could be summarized.

Sentences are long and complex, reducing readability. It is recommended to use clear, concise, and simple language.

Overall, the manuscript addresses an important and practical topic. With the suggested revisions, it could become clearer, more readable, and scientifically stronger, enhancing its suitability for publication.

Sincerely,

dr. matin mojaveri samak

6. PLOS authors have the option to publish the peer review history of their article (what does this mean?). If published, this will include your full peer review and any attached files.

Reviewer #1: No

Reviewer #2: **Yes:**matin mojaveri samak

---

## [Author Response · Author response to Decision Letter 1]

11 Apr 2026

Review Comments to the Author

Reviewer #1:

Dear Reviewer. The authors would like to express our sincere gratitude to you for kindly agreeing to review our manuscript. We are confident that your insightful suggestions will significantly enhance the quality of the work.

All modifications made in response to your comments have been highlighted in red within the revised manuscript

1. The author indicate why the specific focus on teachers was made in this study.This can make the introduction more stronger .

Thanks for your contribution. The authors believe that this suggestion is missing the negation, possibly due to an oversight. Nevertheless, we have reviewed your valuable suggestion and have included it in the manuscript (LINES 133-135) “…Teachers, as directly responsible for supervision during this extensive school period, constitute the key professional group to ensure the safety and adequate care of these students, acting as first responders in emergency situations [29,30].”

2. Non-probability sampling can pose certain limitations in its implementation, as the results may not be fully generalizable to the entire population of teachers. Explain the challenges of this type of sampling and its impact on the final findings.

We appreciate your comment for its accuracy. In our study, we opted for a non probabilistic consecutive sampling method, including all teachers from the Public Early Childhood and Primary Education Centres in the municipality of San Bartolomé (n = 264), which can be considered an approximation to a census type sampling within the studied context. The final sample consisted of 126 participants, due to the voluntary non participation of part of the eligible population. This loss may introduce selection bias and limit the representativeness of the sample; for this reason, the results have been interpreted with caution. As you can see in the limitations section, this observation has been incorporated there (LINE 545-546).

3. Further explanation should be given about how the schools were selected, whether they were randomly selected or purposefully selected for a specific reason? Was it random sampling or not?

Thank you for your observation. As we indicated in the previous point, the educational centres were not selected at random; rather, all available centres in that municipality were included (publics). Therefore, this was not a random sampling procedure. This clarification has been included in the manuscript LINES 163–164 “The sample was selected through non-probabilistic convenience sampling”

4. Are specific statistical analyses considered to compare different groups of teachers (by gender, experience, type of teaching, etc.)?

Thank you for your valuable feedback. Although a table presenting the descriptive statistics of the independent variables (sex, years of experience with students with T1D, teaching in primary or infant education, educational centers) was not included to prioritize the most relevant tables for the manuscript, these characteristics have been described in text at the beginning of the Results section. Educational centers were provided with their own descriptive analysis table (Table 3).

To explore differences between the three aforementioned independent variables and INAPAD results, Table 9 presents the correlation analysis between years of experience and INAPAD score using "Spearman rho," including Figure 1 which graphically represents these two variables. Additionally, sex and INAPAD results were explored through the Mann-Whitney test (Table 10), and the different educational centers (ECPEC) with INAPAD results were analyzed using the Kruskal-Wallis test.

5. If possible, it may be useful to mention challenges or time constraints, as research during this time frame may be affected by factors such as holidays or unexpected circumstances (e.g. health or natural crises)

Your insightful suggestions regarding the “Limitations” section have been incorporated into the manuscript (Line 550-560). “Likewise, the data collection period may have negatively influenced participation, as it coincided with the period following the Christmas holidays, as well as with possible coverage of teacher substitutions. These circumstances may have contributed to teachers' reluctance to participate, among which could be found the high workload, the limited confidence in health-related knowledge, the concern over potential evaluation, as well as the professional implications of their possible responses. The perception of insufficient knowledge to manage urgent situations related to diabetes could be explained by the inadequacy or absence of training programs, stemming from the lack of trainers or school nurses responsible for periodically delivering and assessing teachers' knowledge and skills in this context. As noted previously, a structured implementation of school nursing in Spain remains necessary [21] ”

6. It is important to point out the limitations of the study, especially the low participation rate (47.72%), and the fact that the study results cannot be fully generalized to the entire population is very clear and useful.

Your comments have a great value that we are thankful off. Access to the teachers was arranged through the school principal, who was responsible for distributing the questionnaire link among the teaching staff. Fifteen days after the initial dissemination, the authors asked those in charge to resend a participation reminder to the teachers at their respective centres; however, despite these efforts, participation did not increase. This clarification has been incorporated into the manuscript in the Methodology section, in LINES 208–211 “Fifteen days after the initial distribution, the staff responsible for disseminating the survey link at the centre were asked to remind the faculty members of their opportunity to participate in the questionnaire, and the link was subsequently resent to them.”

Moreover, this aspect—namely, the impossibility of generalising the results to the study population—has been included in the Limitations section, LINES 545-546 “ …(n=126), which limits the generalizability of the results to the selected municipality.”

7. Also, pointing out the lack of similar research in this area helps to better understand the existing research problems and the scope for expanding the research.

Explain more about limitations on participation rates: In addition to stating the percentage of participation, you might explain whether factors such as teacher reluctance or time constraints contributed to the low participation. This can help provide more clarity about the reason for the limitations in the results.

Your insightful suggestions regarding the “Limitations” section have been incorporated into the manuscript (Lines 550-560) “Likewise, the data collection period may have negatively influenced participation, as it coincided with the period following the Christmas holidays, as well as with possible coverage of teacher substitutions. These circumstances may have contributed to teachers' reluctance to participate, among which could be found the high workload, the limited confidence in health-related knowledge, the concern over potential evaluation, as well as the professional implications of their possible responses. The perception of insufficient knowledge to manage urgent situations related to diabetes could be explained by the inadequacy or absence of training programs, stemming from the lack of trainers or school nurses responsible for periodically delivering and assessing teachers' knowledge and skills in this context. As noted previously, a structured implementation of school nursing in Spain remains necessary [21] ”

8. In the section on suggestions for future research, it would be useful to mention the possibility of conducting longitudinal studies rather than just cross-sectional ones.This longitudinal study could help identify changes in teachers' attitudes over time.

9. Perhaps adding details about how this tool could be applied to other schools or different regions could make future research more engaging and scalable.

Thank you for your contribution. In response to your suggestions regarding points 8 and 9, we have incorporated a unified revision into the manuscript. Lines 575-579 now read as follows: “Therefore, it would be appropriate to continue pursuing this line of research through new studies that examine the validity of the instrument across a variety of school settings, including public, subsidised, and private institutions located in different geographical areas. Moreover, conducting longitudinal studies would make it possible to identify changes in teachers’ attitudes over time.”

10. Also, in the section referring to the nurse-to-student ratio, explaining more about the current situation in schools (what this ratio is and what challenges exist) could make the suggestions more effective.

Very relevant contribution, thank you. In this autonomous region (Canary Islands-Spain), there is no official "school nurse" specialty; it is a non-official postgraduate training (master's degree). However, having this training represents a professional boost for graduate nurses. On the other hand, there are official trainings in "Family and Community Nursing" and "Pediatrics" that include these competencies. Shortly after completing this study (October 2025), the Spanish State Gazette published the approval of the Master's degree in Advanced Practice Nursing in School Health.

Therefore, as extensively discussed in the Introduction section (LINES 98-129), school nursing is currently not structurally implemented, although progress is being made in this direction. A new paragraph and reference have been included in the manuscript at LINES 110-113: "It is necessary to mention that school nursing lacks national legal regulation (non-structural implementation), an aspect that may generate inequalities across different regions of Spain [21]. However, in October 2025, the Master's Degree in Advanced Practice Nursing in School Health was recognized in the Official State Gazette [22]"

Therefore, in present, school nursing is not structurally implemented, although progress is being made in this direction. This has been included in the manuscript, LINES 513-516 . “Nevertheless, it must be remembered that school nursing is not structurally established, Nevertheless, it must be remembered that school nursing is not structurally established, which may explain low school nurse-to-student ratio (1:11,000) —far below international recommendations (1:750)—, which reflects a clearly insufficient provision of school nursing services that may be modulating the results of this study [17,19,20].”

11. Explain more about limitations on participation rates: In addition to stating the percentage of participation, you might explain whether factors such as teacher reluctance or time constraints contributed to the low participation. This can help provide more clarity about the reason for the limitations in the results.

We regret to inform you that we believe an error may have occurred, as this comment appears to be duplicated; it is already included under item number 8.

12. In the section on suggestions for future research, it would be useful to mention the possibility of conducting longitudinal studies rather than just cross-sectional ones. This longitudinal study could help identify changes in teachers' attitudes over time.

We regret to inform you that we believe an error may have occurred, as this comment appears to be duplicated; it is already included under item number 7

13. Perhaps adding details about how this tool could be applied to other schools or different regions could make future research more engaging and scalable.

We regret to inform you that we believe an error may have occurred, as this comment appears to be duplicated; it is already included under item number 9.

14. Also, in the section referring to the nurse-to-student ratio, explaining more about the current situation in schools (what this ratio is and what challenges exist) could make the suggestions more effective.

We regret to inform you that we believe an error may have occurred, as this comment appears to be duplicated; it is already included under item number 10.

15. Explain more about limitations on participation rates: In addition to stating the percentage of participation, you might explain whether factors such as teacher reluctance or time constraints contributed to the low participation. This can help provide more clarity about the reason for the limitations in the results.

We regret to inform you that we believe an error may have occurred, as this comment appears to be duplicated; it is already included under item number 11.

16. In the section on suggestions for future research, it would be useful to mention the possibility of conducting longitudinal studies rather than just cross-sectional ones. This longitudinal study could help identify changes in teachers' attitudes over time.

We regret to inform you that we believe an error may have occurred, as this comment appears to be duplicated; it is already included under item number 7.

17. Perhaps adding details about how this tool could be applied to other schools or different regions could make future research more engaging and scalable.

We regret to inform you that we believe an error may have occurred, as this comment appears to be duplicated; it is already included under item number 9 & 13.

18. Also, in the section referring to the nurse-to-student ratio, explaining more about the current situation in schools (what this ratio is and what challenges exist) could make the suggestions more effective.

We regret to inform you that we believe an error may have occurred, as this comment appears to be duplicated; it is already included under item numbers 10 & 14.

19. Mentioning the need for simulations and evaluating educational impacts can guide future research to examine whether these trainings have a positive impact on teacher preparation.

20. It may be useful to refer to methods for evaluating the impact of training. For example, do teachers score better on subsequent assessments after training? This can strengthen conclusions and emphasize the effectiveness of training programs in particular.

21. In the case of emergency training and simulations, more details may be provided on how these simulations will be implemented (e.g., use of medical simulation technologies or training workshops)

The reviewer is correct; this information is highly relevant to the manuscript. Thank you very much.

Since suggestions nos. 19, 20, and 21 are interrelated, they have been unified to provide a cohesive response and redaction in the Discussion section. On LINES 561-571 the following has been included:

“Among the strategies to enhance teachers' knowledge and confidence in caring for students with diabetes, theoretical training stands out, as certain online programs have demonstrated significant improvements in both knowledge and teacher confidence, with retention maintained up to 12 months. Likewise, evaluating training impact is equally advisable to assess program effectiveness. Authors recommend pre-post evaluations. Thus, a 2026 study found significant increases in knowledge, perceived self-efficacy, and ability to recognize hypoglycemia symptoms (all p<0.001), similar to virtual pilot studies showing significant improvements in diabetes technology use, basic management, and ketone handling (all p<0.001). Finally, simulation emerges as the most robust evidence, with numerous studies demonstrating significantly increased confidence in managing these patients during emergency situations [41-43].”

Reviewer #2:

The authors would like to express our sincere gratitude to you for kindly agreeing to review our manuscript. We are confident that your insightful suggestions will significantly enhance the quality of the work.

All modifications made in response to your comments have been highlighted in blue within the revised manuscript

1. Use of Technical Terms: In the introduction and abstract, the use of terms such as INAPAD-18 is appropriate. However, it may be confusing for non-specialist readers. I recommend including a

---

## [Editor Report · Decision Letter 1]

19 May 2026

Preschool and Primary School Teachers’ Attitudes Toward Students with Type 1 Diabetes: A Cross-Sectional Study

PONE-D-25-62219R1

Dear Dr. Díaz-González,

We’re pleased to inform you that your manuscript has been judged scientifically suitable for publication and will be formally accepted for publication once it meets all outstanding technical requirements.

Kind regards,

Afagh Hassanzadeh Rad

Academic Editor

PLOS One
---

## [Editor Report · Acceptance letter]

PONE-D-25-62219R1

PLOS One

Dear Dr. Díaz-González,

I'm pleased to inform you that your manuscript has been deemed suitable for publication in PLOS One. Congratulations! Your manuscript is now being handed over to our production team.

Kind regards,

on behalf of

Dr. Afagh Hassanzadeh Rad

Academic Editor

PLOS One